# 🌳 OPENCARBONEVAL: HOW MUCH $CO_2$ WILL YOUR LARGE MODEL EXHALE IN TRAINING PROCESS?

## ABSTRACT

Data, model and hardware are crucial components in the development of large scale machine learning models. The training of such models necessitates substantial computational resources, energy consumption, and raw materials, resulting in significant environmental implications. However, the environmental impact of these models has been largely overlooked due to a lack of assessment and analysis of their carbon footprint. In this paper, we present OpenCarbonEval, a carbon emission estimation tool to quantify the environmental implications of large scale machine learning models given their total training computations and hardware configurations. In OpenCarbonEval, we conducted a comprehensive dynamic analysis of the interrelationships among data, models, and hardware throughout the model training process, aiming to forecast the carbon emission of large scale models more accurately. We validated our approach on real-world dataset, and experimental results demonstrate that OpenCarbonEval can predict energy costs and carbon emissions more accurately than previous methods. Furthermore, it can be seamlessly applied to various machine learning tasks without a precision decline. By quantifying the environmental impact of large-scale models, OpenCarbonEval promotes sustainable AI development and deployment, contributing to a more environmentally responsible future for the AI community.

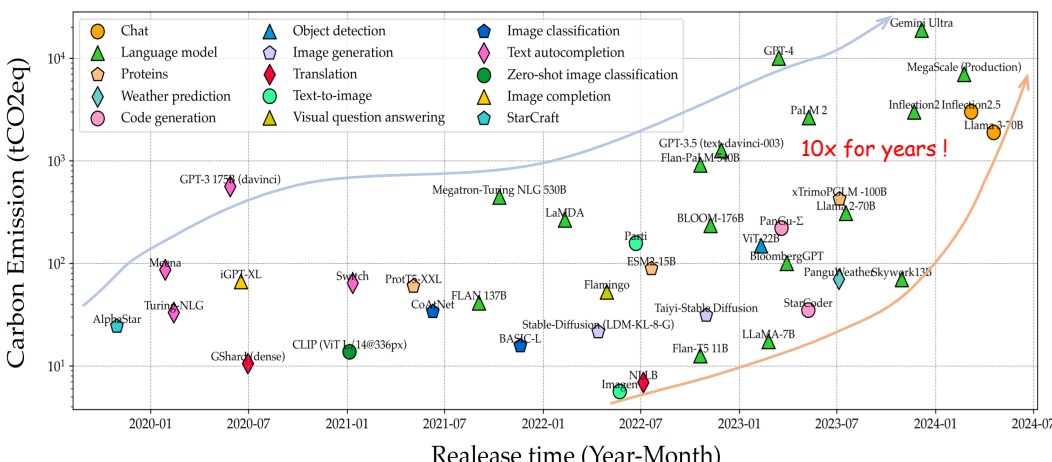

Figure 1: Large-scale models' environmental impact covering 42 large-scale AI models across 15 tasks. The carbon footprint of large-scale ML models has significantly increased over time, with annual growth rates exceeding tenfold. A detailed analysis is in Section 4.5
.

## 1 INTRODUCTION

Recently, large scale ML models like Large Language Models (LLMs) (OpenAI, 2023) and Multi-modal Large Language Models (MLLMs) (Chen et al., 2023) have exhibited remarkable intelligence across a wide range of tasks, largely attributed to the advancement of their scaling laws (Henighan et al., 2020; Kaplan et al., 2020; Zhai et al., 2022). However, as the scale of model parameters and training sets increases, the computational overhead of training and maintaining large-scale models becomes exorbitantly huge, resulting in significant environmental impacts. For instance, training a GPT-3 (Brown et al., 2020) with 175B parameters will consume nearly 1300MWh of electricity (Patterson et al., 2021), roughly equivalent to the annual electricity consumption of 130 households in the US. Meanwhile, its corresponding *carbon dioxide equivalent* (CO2eq) is about 552 tons (Patterson et al., 2021), which is three times the CO2eq emissions of jet plane round trip between San Francisco and New York. Therefore, the ML community should pay greater attention to the energy consumption and environmental impact of these large-scale ML models.

Previous works, such as MLCO2 (Lacoste et al., 2019) and GreenAlgorithm (Lannelongue et al., 2021), have proposed to calculate the carbon emission of ML tasks based on some key parameters like GPU usage, training duration, and data center efficiency. These methods heavily rely on exact information about the training process, implying that only model developers can use these tools to estimate the energy consumption and carbon emissions of their trained models. To break away from this limitation, LLMCarbon (Faiz et al., 2023) presents an end-to-end approach for estimating carbon emissions before model training. It inputs the key architecture parameters of LLM into its specially designed FLOP-model and efficiency-model, which can be used to predict the training duration and carbon emission. However, the key steps of this method are all designed for LLM, and the polynomial fitting coefficients in its efficiency-model are completely unsuitable for other ML tasks, *e.g.* image generation. Furthermore, these estimation methods often assume a static or average workload, failing to capture the dynamic nature of the training process of large scale ML models. This oversight can lead to significant inaccuracies in energy consumption and carbon emission estimates.

To ensure a comprehensive comparison and analysis for the energy and carbon footprint of various past and future large-scale models, we have identified key challenges: an accurate and transparent anticipatory approach is needed, which can use basic information of training to predict energy consumption and carbon emissions accurately. This approach should also produce fair and comparable results across diverse ML tasks in various fields.

In this paper, we propose **OpenCarbonEval**, a carbon emission estimation tool to quantify the environmental implications of large scale ML models given their total training computations and hardware configurations. In OpenCarbonEval, our contributions are summarized as follows:

- **A Carbon Emission Estimation Method for Various ML Tasks** We propose a novel method to accurately estimate the dynamic power consumption and carbon emission of large scale ML models across various ML tasks, using two basic information including training computation and hardware configuration.

- **The first Open Source Dataset about the Carbon Footprint of Large Scale ML Models** We collect and open source the OpenCarbonEval dataset comprising 110 real-world data of large scale ML models across 20 ML tasks on their carbon footprint.

- **Empirical Validation on the Method of Carbon Emission Estimation** We conduct a statistical analysis of the benefits and limitations of carbon emission methods, providing valuable insights for future research.

The results of our analysis demonstrate that the predictions generated by OpenCarbonEval exhibit a high accuracy with real-world data, enabling us to produce more accurate predictions for various ML Tasks. Furthermore, to promote a more transparent and sustainable ML community, we will open source all the OpenCarbonEval dataset and the estimation tools used.

## 2 RELATED WORK

Over the past decade, deep learning has experienced remarkable advancements, particularly with the recent dominance of large-scale models. These models have significantly increased in model size

and training data (Villalobos et al., 2024). While their performance has improved dramatically, the computational costs have grown exponentially (Sevilla et al., 2022). This surge in computational demand results in substantial energy consumption, leading to considerable greenhouse gas emissions. As we continue to develop more and larger AI models in the foreseeable future, understanding their energy costs and environmental impact becomes crucial.

Previous works (Wu et al., 2022; Luccioni et al., 2023) usually divide the carbon footprint of AI models into two parts: operational carbon and embodied carbon. Operational carbon includes the carbon emissions generated by producing the electricity required for training an AI model and using it for inference on computing devices. Embodied carbon means the equivalent carbon emissions from manufacturing the computing devices. During the training phase of large models, the primary contributor to carbon emissions is operational carbon, which results indirectly from the energy consumption of AI computing chips. It can be calculated by multiplying the energy cost for AI computing $E(kWh)$ by the regional carbon intensity $I(kgCO2eq/kWh)$.

Related works have proposed some methods for calculating the energy cost and carbon footprint of training AI models, we can broadly categorize them into three types:

**Retrospective Calculation Method**: MLCO2 (Lacoste et al., 2019) and GreenAlgorithm (Lannelongue et al., 2021) can estimate the energy consumption and carbon footprint of ML tasks based on user input information such as device type, training duration, and power grid area. The difference is that the latter accounts for additional CPU and memory consumption. Although these inputs are independent of the model, their application is significantly limited. This is because, aside from model developers, others may not have access to the exact training duration. Consequently, we can not apply these estimates to models that have not been trained or those without reported training duration.

**Real-time Monitoring Method**: CodeCarbon (Courty et al., 2024), Carbontracker (Anthony et al., 2020), and Eco2AI (Budennyy et al., 2022) are designed to run in parallel with ML tasks for real-time monitoring. Each provides a Python library that can be integrated into existing training scripts to capture dynamic hardware energy consumption throughout the process. While this approach is theoretically precise, its intrusive nature or lack of integration with existing distributed training frameworks may limit widespread adoption. This method also cannot be used to analyze existing models or predict future models' carbon emissions.

**Anticipatory Estimation Method**: LLMCarbon (Faiz et al., 2023) is first end-to-end approach for estimating model carbon emissions before training. It is specifically designed for LLM architectures, which includes a FLOP-model to estimate total computation and an efficiency-model to estimate average hardware computation speed. By combining them, this anticipatory method can predict training time and carbon footprint based on the model's key information before training. However, since the FLOP-model and efficiency-model are tailored to LLM frameworks, the polynomial coefficients used in the method are difficult to apply to other hardware types or task architectures.

Our OpenCarbonEval is also an anticipatory method, this general framework leverages existing model training statistics and approximates the dynamic computation processes of hardware, which can predict training times for AI models across various architectures and tasks. This enables fair and comparable estimate results of energy consumption and carbon footprint.

# 3 OPENCARBONEVAL

Building on previous research (Faiz et al., 2023; Luccioni et al., 2023), we categorize the overall carbon emissions during the training process of ML models into two main components: operational carbon emissions from energy consumption and embodied carbon emissions associated with the materials and processes involved in hardware production.

## 3.1 OPERATIONAL CARBON

Operational carbon, produced by generating the electricity necessary for powering model training, is a significant component of the environmental impact associated with machine learning and artificial intelligence systems. This type of carbon emission arises from the energy consumption required to run the computational processes involved in training ML models, which could be calculated as:

$$C_{\text{operational}} = E \cdot I \tag{1}$$

where $C_{\text{operational}}$ indicates the amount of emitted carbon dioxide ($kgCO2eq$), $E$ ($kWh$) indicates the energy consumed for model training and $I(kgCO2eq/kWh)$ indicates the emitted $CO_2$ per $kWh$ energy consumed.

## 3.2 Dynamic Power Consumption

In Eq. (1), the grid's carbon intensity $I$ is a coefficient ($kgCO2eq/kWh$) depends on the electricity source that powers training process which is often related to the region where the data center is located. The energy consumption $E$ is often calculated by multiplying the number of GPU hours used by the thermal design power (TDP) of those GPUs and the carbon intensity ($I$) of the energy grid used to power the hardware, which can be written as follows:

$$E = TDP \cdot T_{\text{train}} \cdot N_{\text{GPU}} \tag{2}$$

where $T_{\text{train}}$ indicate the training time of the model and $N_{\text{GPU}}$ is the number of all hardware involved in training process. In Eq. (2), $TDP$ and $N_{\text{GPU}}$ are typically constants that are independent of time. Therefore, we mainly study the energy consumption over the training time $T_{\text{train}}$ in this section.

**Little's Law in training process** In the training process of a ML model, the hardware initially loads the model and data from memory. This process then rapidly transitions to a steady state for efficient processing, analogous to a queuing system. In the early stages of a queuing system, when the queue is empty, no waiting is necessary. However, once the queue reaches capacity, subsequent data must wait in line. This waiting period effectively constitutes the training time, denoted as $T_{\text{train}}$. Therefore, we simulate the queuing process and use Little's Law Little & Graves (2008) to model the relationship between total computation, training speed and GPU time during the model training process. Consider a short interval $(t, t + \Delta t)$ within the training time $T_{\text{train}}$, we can get a product relationship from little's law as follows:

$$L_{\Delta t} = \bar{\lambda} \cdot \Delta t \tag{3}$$

where $L_{\Delta t}$ is the total computation processed by GPUs and $\bar{\lambda}$ is the average training speed during $\Delta t$. In our approach, we divide $T_{\text{train}}$ into the same $n$ parts and use $\Delta t_i = \Delta t$ and $\bar{\lambda}_i$ to denote the i-th time interval and the average speed. By adding up all the time intervals according to Eq. (3), we have

$$L_{\text{computation}} = \sum_{i=0}^{n} \bar{\lambda}_i \cdot \Delta t \tag{4}$$

However, it is not straightforward to calculate their average speed $\bar{\lambda}_i$ for all $\Delta t_i$. Hence, we calculate the form of formula 4 when $\Delta t \to 0$, where the average speed $\bar{\lambda}_i$ is an instantaneous speed that changes over time $f(t)$. This process can be written as:

$$L_{\text{computation}} = \int_0^{T_{\text{train}}} f(t)dt \tag{5}$$

From Eq. (5), we can solve for the training time $T_{\text{train}}$ and bring it into Eq. (1) to obtain operational carbon if $f(t)$ is available. However, the train speed $f(t)$ is often difficult to estimate due to different hardware configurations and training setups. Therefore, we focus on the selection of $f(t)$ and validate its effectiveness in the following sections of this paper.

**The inspiration from real-world training process** In the training process of an ML model, the hardware initially loads the model and data from memory. Subsequently, the hardware quickly reaches a steady state, efficiently processing the gradients and other tensors generated during model training. To simulate this process, the function $f(t)$ we choose should satisfy the requirement of quickly entering a relatively stable state, which could be expressed as follows:

$$\lim_{t \to \infty} f'(t) = 0 \tag{6}$$

**The challenge of insufficient data** After identifying the general trend of $f(t)$, we need to determine the parameters in $f(t)$ based on real-world data points. However, there is limited discussion within

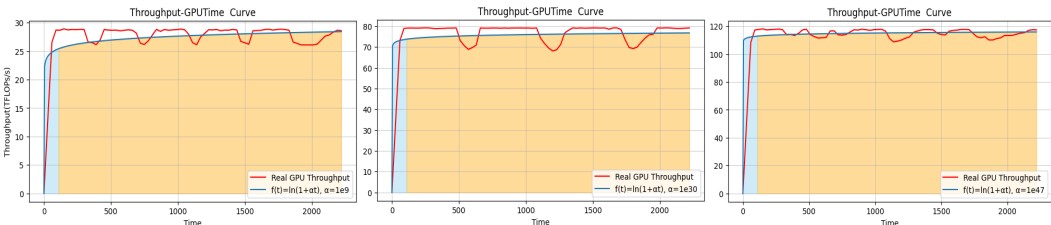

Figure 2: The comparison between real-world training speed and $f(t) = ln(1 + \alpha t)$ under different training setting. More detailed analysis of $\alpha$ in shown in Section 4.2.

the open-source community regarding the training details and carbon footprint of large-scale ML models, making it difficult to find enough real-world data to fit $f(t)$. Therefore, due to the lack of enough real-world data, we could not set too many parameters in $f(t)$.

Combining the above two considerations, our $f(t)$ is formulated as follows:

$$f(t) = ln(1 + \alpha t) \tag{7}$$

where only one parameter $\alpha$ is used to determine the shape of $f(t)$. As shown in Fig. 2, we conducted experiments under various settings and compared the results with the function $f(t) = ln(1 + \alpha t)$. To reflect the correlation between different values of $\alpha$ and the hardware, we collect the avaliable data of all large-scale machine learning models from EpochAI as of August 2024, totaling 110 examples. We will open-source the data we used to the community. A detailed discussion of the findings from these data is provided in Section 4.

### 3.3 EMBODIED CARBON

Embodied carbon, representing the emission associated with hardware manufacturing and the processes involved in producing given hardware. While the production of these emissions is exclusively limited to the manufacturing process, this total amount is usually spread over the time during which equipment is used by dividing the total embodied emissions by the time of use. this process can be calculated as follows:

$$C_{\text{embodied}} = \frac{C_{\text{lifelong}}}{T_{\text{lifelong}}} \cdot T_{\text{train}} \cdot N_{\text{GPU}} \tag{8}$$

where $C_{\text{embodied}}$ and $T_{\text{train}}$ indicate the embodied carbon and training time of the model to be estimated respectively, $C_{\text{lifelong}}$ and $T_{\text{lifelong}}$ represent the product carbon and life time of the GPUs respectively, and $N_{\text{GPU}}$ is the number of all hardware involved in training process. With other information already known, the key to predicting $C_{\text{embodied}}$ becomes similar to that of $C_{\text{operational}}$, *i.e.*, predict the training time $T_{\text{train}}$.

## 4 VALIDATION

### 4.1 DATASET

**OpenCarbonEval Dataset** We collect the key parameters from EpochAI's "Notable AI Models" dataset[1], including *Training Compute*, *Training Time*, *Training Hardware* and *Hardware Quantity*. For over 800 entries in EpochAI, we drop the *null* value and keep 110 records to obtain the statistical information for our method. The remaining dataset encompasses 20 ML tasks and the majority of common model frameworks, such as LLMs, vision, image generation, multimodal, speech, and video. It also covers 26 different hardware devices, *e.g.* NVIDIA V100, A100, and Google TPU v4 and so on. We estimate the $\alpha$ parameter for all records, allowing users to select the $\alpha$ value from a record with similar configurations to their model, or use the mean value for their hardware type, as we do. Using the statistical information from the dataset and OpenCarbonEval estimation method, we can predict the carbon emissions for any ML model that provides the total computation and hardware

---

[1]https://epochai.org/data/notable-ai-models?view=table

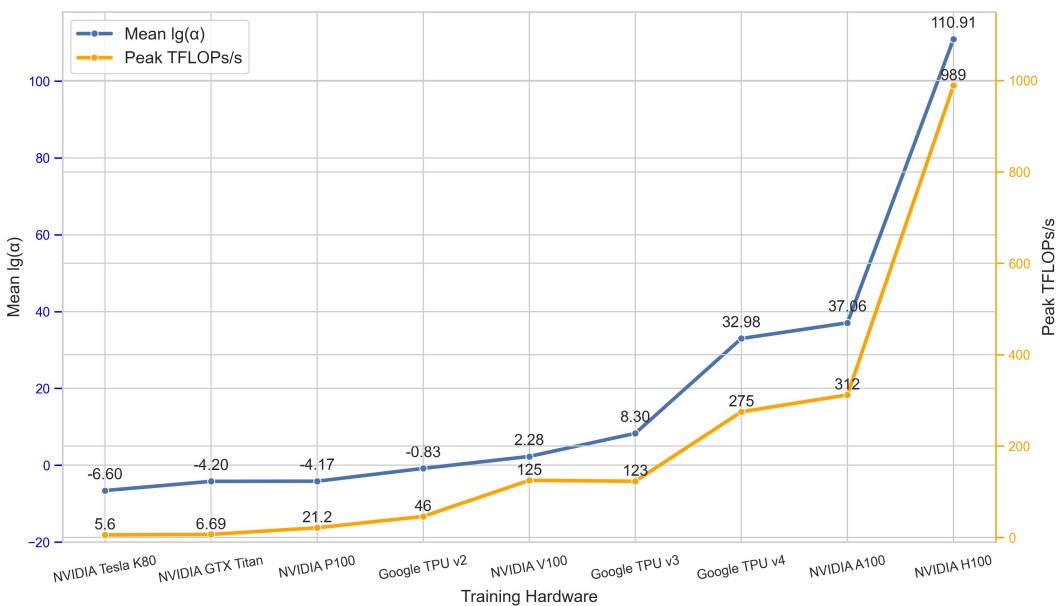

Figure 3: Comparison between mean estimated $\alpha$ values(blue color) and theoretical peak speeds in real world(orange color) for different hardware devices. It shows the high consistency in the trends of the two values across different devices, which validates the effectiveness of using the $\alpha$ parameter to model computation speed. Note that peak speed information is invisible in our model, and the peak speed is typically unattainable during the training process.

type. In our experiments, we use *Training Time* for $T_{\text{train}}$, and *Training Compute* for $L_{\text{computation}}$ and *Hardware Quantity* for $N_{\text{GPU}}$. By substituting $T_{\text{train}}$ and $L_{\text{computation}}$ into Eq. (5), we can estimate the value of the $\alpha$ parameter in $f(t)$.

**Evaluation Set** We curated a diverse evaluation set of open-source large-scale models, varying in functionality, input data, geographical region, and computing device used for training to serve as test data points. We present results from an array of open-sourced LLMs, such as **ChatGLM** Zeng et al. (2022) with 130 billion parameters, **BLOOM** (Workshop et al., 2022) with 176 billion parameters, **StarCoder** (Li et al., 2023), a generative model for code synthesis and **LLaMa-3-70B** (AI@Meta, 2024), a model trained on Meta's large-scale AI clusters which takes data and scale to new heights. While the scaling laws of language models have been well-established, those of visual models remain an active area of exploration, with a notable absence of carbon emission predictions for this type of model. So we also add two iconic models, Vision Transformer (**ViT-L/16**) (Dosovitskiy et al., 2020) and Swin Transformer (**Swin-L**) (Liu et al., 2021) into our validation.

### 4.2 THE IMPACT OF HARDWARE

To investigate the impact of hardware, we first extract the total computation $L_{\text{computation}}$ and training time $T_{\text{train}}$ from our OpenCarbonEval dataset. Subsequently, we bring them to Eq. (4) and Eq. (7) to obtain the value of $\alpha$ for each large scale ML model that is categorize by the training hardware.

**The values of $\alpha$ exhibit a similar upward trend to the real-world hardware training speed, indicating a positive correlation** To demonstrate the correlation between the parameter $\alpha$ and the real-world hardware performance, we compared the mean estimated $\alpha$ values from different devices with their theoretical peak speeds. As illustrated in Fig. 3, the values of $\alpha$ naturally exhibit the same trend with the hardware peak training speed (TFLOPs/s). It indicates that $\alpha$ values can show significant discrepancies due to differences in GPU performance, *i.e.*, devices with better actual performance will have larger estimated $\alpha$ values. This further validates the effectiveness of the function form $f(t)$ and the parameter $\alpha$, and demonstrates their potential to adapt to future advancements in computing hardware.

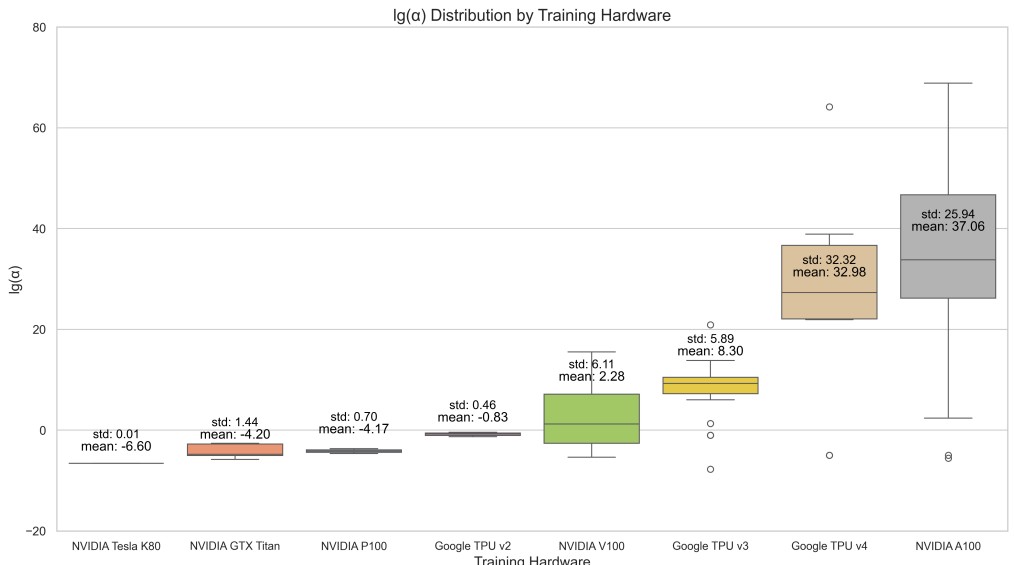

Figure 4: The $\alpha$ distribution by different training hardware. We estimated the parameter $\alpha$ values for each record in the dataset and conducted statistical analysis based on hardware types. The $\alpha$ values differ significantly across different hardware categories. Within each hardware, the range of $\alpha$ values also varies, reflecting the diversity of real-world samples. Hardware types with only one record have been omitted in this figure.

**The value of $\alpha$ is predominantly determined by the specific training hardware.** As illustrated in Figure Fig. 4, Different types of hardware often exhibit distinct alpha ranges, which can vary significantly based on their architectural and design characteristics. However, when the computing power of the hardware is comparable, these alpha ranges tend to overlap *e.g.* TPUv4 and NVIDIA A100, indicating a convergence in performance metrics despite the underlying differences. For the purpose of facilitating analysis, we hereafter utilize the mean $\alpha$ values for each hardware type, as presented in Fig. 4, to compute the energy consumption and carbon emissions of various ML models.

### 4.3 OPERATIONAL CARBON FOOTPRINT VALIDATION

Table 1 presents the result of OpenCarbonEval on various large-scale models. We have compiled a comprehensive table that outlines all the parameters necessary for carbon emission estimation. Within this table, ZettaFLOPs represents the total computation amount required for effective model training, parameter represents the number of model parameter and $I(gCO2eq/kWh)$ represents the carbon intensity in Eq. (1). From the Table 1, we have the following observations:

**Compared with LLMCarbon, OpenCarbonEval exhibits a significantly lower relative error in predicting carbon emissions across different compute devices.** In contrast to the actual CO2eq emissions, LLMCarbon exhibited significant errors, with a notable discrepancy of up to 114.5% in predicting the LLaMa-3's carbon footprint. This is attributed to its modeling approach not being transferable to new GPUs. In contrast, OpenCarbonEval demonstrates remarkable accuracy, with small relative errors at all test data points, thereby validating its effectiveness.

**OpenCarbonEval consistently achieves low relative errors in its predictions for both visual and language models, demonstrating its versatility and robustness across different modalities.** Notably, when predicting the carbon footprint of visual models such as ViT/16-L and Swin-L, OpenCarbonEval still outperforms LLMCarbon, achieving relatively accurate predictions. This superiority can be attributed to OpenCarbonEval's unique strength in establishing a unified task set that can accommodate all modalities. The error rate on ViT-L/16 may be mainly attributed to the significant differences in TPUv3 types or abnormal data in our dataset. We believe this result can be further improved by more available open source data.

Table 1: Operational carbon of various models on different GPU. The result of the best method is bolded. Error rate represents the relative error between the predicted value and the actual value. We use the self-reported results whenever available.

| Method | GLM | BLOOM | StarCoder | LLaMa-3 | ViT-L/16 | Swin-L |
|---|---|---|---|---|---|---|
| Params | 130B | 176B | 15B | 70B | 307M | 197M |
| ZettaFLOPs | 312 | 387 | 93 | 6300 | 0.53 | 0.40 |
| Hardware | A100 | A100 | A100 | H100 | TPUv3 | V100 |
| $I$ ($gCO2eq/kWh$) | 581 | 57 | 155 | 424 | 369 | 369 |
| Actual CO2eq ($t$) | 257 | 24.7 | 17.26 | 1900 | 2.71 | 0.80 |
| LLMCarbon | 153.11 | 19.89 | 14.14 | 4074.63 | 0.20 | 0.10 |
| Error Rate | -40.4% | -19.4% | -18.1% | +114.5% | -92.6% | -87.5% |
| OpenCarbonEval | **189.75** | **23.04** | **15.26** | **1866.90** | **0.39** | **0.59** |
| Error Rate | **-26.1%** | **-6.7%** | **-11.6%** | **-1.7%** | **-85.5%** | **-26.8%** |

Table 2: Different embodied carbon prediction results on various models by OpenCarbonEval. We use the self-reported results whenever available.

| | GLM | BLOOM | StarCoder | LLaMa-3 | ViT-L/16 | Swin-L |
|---|---|---|---|---|---|---|
| Hardware Type | A100 | A100 | A100 | H100 | TPUv3 | V100 |
| TSMC process | 7 nm | 7 nm | 7 nm | 4 nm | 16 nm | 12 nm |
| Die Size | 826 $mm^2$ | 826 $mm^2$ | 826 $mm^2$ | 814 $mm^2$ | 700 $mm^2$ | 815 $mm^2$ |
| $C_{\text{lifelong}}/T_{\text{lifelong}}$ | 1.5 | 1.5 | 1.5 | 1.7 | 0.8 | 1.1 |
| Actual embodied $CO_2$eq (kg) | 1634.50 | 1631.23 | 480.38 | 10880.0 | 13.06 | 7.92 |
| LLMCarbon | 898.37 | 1090.65 | 285.23 | 21211.80 | 0.88 | 0.91 |
| Error Rate | -45.0% | -33.1% | -40.6% | +95.0% | -93.3% | -88.5% |
| OpenCarbonEval | **1224.75** | **1516.11** | **369.24** | **10693.14** | **3.40** | **5.82** |
| Error rate | **-25.1%** | **-7.1%** | **-23.1%** | **-1.7%** | **-74.0%** | **-26.5%** |

## 4.4 Embodied Carbon Footprint Validation

By reviewing LLMCarbon and obtaining specifications for different types of hardware materials, we calculated the embodied carbon footprint using Eq. (8), assuming a 1-year effective lifespan for each hardware component. This approach allows us to account for the embodied carbon emissions resulting from the manufacturing process, which is an essential aspect of comprehensive carbon evaluation. As shown in Table 2, although embodied carbon constitutes a relatively small proportion of the total carbon evaluation, OpenCarbonEval can still maintain high prediction accuracy, demonstrating the effectiveness of our approach approach in estimating carbon emissions for large scale ML model.

## 4.5 Case Study

In this section, we conduct a carbon footprint estimation for 42 models across 15 different tasks, using necessary information such as total computation, hardware type, and training location, which correspond to the $L_{\text{computation}}$, $f(t)$ with parameter $\alpha$, and carbon intensity $I$ in our method. As illustrated in Fig. 1, the carbon footprint of large-scale ML models has significantly increased over time, with annual growth rates exceeding tenfold. While language models (LLMs) remain the largest contributors to carbon emissions, other models such as image generation and visual question answering (VQA) are also adding to this escalating impact. Consequently, a comprehensive framework like OpenCarbonEval, which uniformly assesses all ML tasks and devices, is crucial for advancing sustainability in the AI community.

## 5 DISCUSSION AND LIMITATIONS

**Insufficient Real-world Data** While OpenCarbonEval provides a unified framework for estimating carbon emissions, it is not without its limitations. For example, due to the limited availability of real-world data on ML models, significant deviations in predictions are possible in some scenarios. Besides, various training setups, such as deep learning frameworks and distributed parallel strategies, can significantly impact training speeds and duration. However, the current scarcity of real-world data hinders a comprehensive analysis of these factors. So our framework is not primarily focused on accuracy, but rather on the predictability and universality it provides. For AI developers who require precise values of their model's energy consumption and carbon emissions, we recommend using the real-time monitoring methods (Courty et al., 2024) mentioned in Section 2 to obtain more reliable results and report their results.

**Carbon Footprint more than GPU** Previous works (Lannelongue et al., 2021; Faiz et al., 2023) assumed that devices such as CPUs and memories operated at constant power, and incorporated the energy consumption caused by these devices into Eq. (2) to account for the carbon footprint brought by additional equipment. Alternatively, from the perspective of the data center (Wu et al., 2022), the energy consumption generated by GPUs can be multiplied by Power Usage Effectiveness (PUE), to obtain the overall energy consumption. Both of them are feasible solutions, but due to the ground truth of the carbon footprint model in real-world data is mainly calculated by GPU consumption, we did not incorporate these additional consumptions into our framework.

**Broder Impact on Environmental Sustainability** The increasing carbon footprint of large-scale AI models has significant implications for the environment and sustainability. Our analysis using OpenCarbonEval reveals a concerning trend of growing carbon emissions associated with the development and deployment of these models. This highlights the need for the AI community to prioritize environmental sustainability alongside performance and efficiency. Furthermore, the environmental impact of AI models can have far-reaching consequences, including contributing to climate change, air pollution, and e-waste generation. By providing a unified framework for predicting carbon emissions, OpenCarbonEval can facilitate the development of more environmentally friendly AI models and encourage responsible AI practices. This includes promoting transparency and accountability in AI development, encouraging sustainable AI design and deployment, and fostering a culture of environmental responsibility within the AI community.

## 6 CONCLUSION

In this paper, we present OpenCarbonEval, a carbon emission estimation tool to quantify the environmental implications of large scale ML models in their training process. OpenCarbonEval is able to accurately estimate the carbon emission and energy consumption of various large scale ML models across various ML tasks, resulting in a more carbon-transparent training process. By leveraging OpenCarbonEval, we collect the first open source carbon footprint dataset comprising the carbon footprint for training large scale ML models. Furthermore, our systematic analysis of the estimation for carbon emissions across various ML tasks provides valuable insights for future research, contributing to the development of more sustainable large scale ML models.

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

APPENDIX

# A  ADDITIONAL INFORMATION OF THE EVALUATED MODEL

The information of evaluated models Fig. 1 are mostly from EpochAI (Epoch AI, 2023).

Figure 1 provides a comprehensive overview of the carbon footprint of large-scale AI models, spanning 42 models across 15 tasks, as systematically classified by EpochAI (Epoch AI, 2023).

**Chat** LLaMa-3-70B (AI@Meta, 2024), Inflection 2.5 [2].

**Language model** Gemini Ultra (Team et al., 2023), MegaScale (Prduction) (Jiang et al., 2024), Inflection 2[1] , GPT-4 (OpenAI, 2023), PaLM-2 (Anil et al., 2023), GPT-3.5, Flan-PaLM 540B, Flan-T5-11B, Flan-137B (Chung et al., 2024), Megatron-Turing NLG 530B (Narayanan et al., 2021), LaMDA (Thoppilan et al., 2022), LLaMa (Touvron et al., 2023),LLaMa-2 (Touvron et al., 2023), BLOOM (Workshop et al., 2022), Skywork-13B (Wei et al., 2023), BloombergGPT (Wu et al., 2023).

**Proteins** ProT5-XXL (Elnaggar et al., 2021), ESM2-15B (Lin et al., 2023), xTrimoPGLM - 100B (Chen et al., 2024).

**Weather prediction** Pangu Weather (Bi et al., 2022).

**Code generation** Pangu-$\Sigma$ (Ren et al., 2023), StarCoder (Li et al., 2023).

**Object detection** ViT-22B  (Dehghani et al., 2023)

**Image generation** Stable Diffusion (LDM-KL-8-G)  (Rombach et al., 2022), Taiyi-Stable Diffusion (Zhang et al., 2022)

**Translation** Gshard (dense) (Lepikhin et al., 2020), NLLB (Costa-jussà et al., 2022)

**Text-to-image** Imagen (Saharia et al., 2022), Parti (Yu et al., 2022b).

**Visual question answering** Flamingo (Alayrac et al., 2022).

**Image classification** Meta Pseudo Label  (Pham et al., 2021), CoAtNet (Dai et al., 2021), CoCa (Yu et al., 2022a), BASIC-L (Pham et al., 2023).

**Text autocompletion** GPT-3-175B (Brown et al., 2020), Turing-NLG (Rajbhandari et al., 2020), Meena (Adiwardana et al., 2020), Switch (Fedus et al., 2022).

**Zero-shot image classification** CLIP (ViT L/14@336px) (Radford et al., 2021).

**Image completion** iGPT-XL (Chen et al., 2020).

**StarCraft** AlphaStar (Vinyals et al., 2019).

---

[2]https://inflection.ai/

