# OpenReview forum: "OpenCarbonEval: How much $CO_2$  will your large model exhale in training process?"
_ICLR.cc/2025/Conference — ICLR 2025 Conference Withdrawn Submission_

### Official Review · Reviewer_NaUT · 2024-10-28

**Soundness:** 1
**Presentation:** 2
**Contribution:** 2
**Rating:** 5
**Confidence:** 1

**Summary:**

The training of machine learning (ML) models significantly contributes to global carbon emissions. This paper introduces OpenCarbonEval, an advanced estimation tool designed to quantify the carbon impact of large-scale ML models based on their total training computations and hardware configurations. The tool's accuracy is validated using real-world datasets, and experimental results demonstrate that OpenCarbonEval provides more precise predictions of energy consumption and carbon emissions than previous approaches.

**Strengths:**

1. The paper works on an important topic.
2. The paper identifies the shortcoming of preivous works (Faiz et al., 2023): the polynomial approximation for the system efficiency and hardware utiliation estimation is not accurate.

**Weaknesses:**

1. **Simplified yet more accurate formulation???**: The functions presented in Equations (3) through (7) lack clarity in their intended function and accuracy. While polynomial approximations may lack precision, Equation (7) is simplified even further than LLMCarbon, containing only a single parameter compared to the multi-parameter nature of polynomial approximations. Why is this single-parameter approach purported to yield higher accuracy? The authors are encouraged to offer detailed explanations or empirical validation demonstrating how and why Equation (7) leads to improved accuracy over traditional polynomial approximations.

2. **Consideration of GPU count and parallelism settings**: The paper does not discuss varying GPU counts in training configurations, appearing to assume a single-GPU setup. It also does not address different training parallelism types, such as data, tensor, pipeline, or expert parallelism, all of which may affect results depending on GPU count. Without incorporating these parallelism factors, it is unclear how OpenCarbonEval achieves greater accuracy. How does this work account for different parallelism strategies, and are there empirical results confirming its accuracy across these configurations? Additionally, Figure 4 lacks context: how many GPUs are represented, why do some GPUs exhibit smaller variance, and how many GPUs are used for training in Tables 1 and 2?

3. **Lack of model architecture information**: The study appears to consider only the number of parameters in ML models, without accounting for architecture specifics. While scaling laws suggest that architecture does not impact model accuracy, it significantly affects training throughput across various architectures (see Megatron paper: https://parsa.epfl.ch/course-info/cs723/papers/Megatron.pdf). The authors should provide empirical evidence to demonstrate that model architecture does not impact the carbon footprint of training.

4. **Dataset limitations**: The dataset used is limited and lacks comprehensive real-world data. Among the 863 entries in the provided table (https://epochai.org/data/notable-ai-models?view=table), only 176 entries include training times, 158 provide GPU counts, and only 31 report hardware utilization, leaving most entries without training times or hardware utilization data. With such limited information, how is \( f(x) \) in Equation (5) trained and validated? Furthermore, 603 of the 863 entries are classified as "likely," "speculative," or "no confidence." Does OpenCarbonEval rely on these uncertain data points for validation while claiming higher accuracy? The authors should discuss the limitations associated with the dataset quality and address the impact on the reliability of their conclusions.

**Questions:**

Please comment on the following weakness:

1. **Simplified yet more accurate formulation???**: The functions presented in Equations (3) through (7) lack clarity in their intended function and accuracy. While polynomial approximations may lack precision, Equation (7) is simplified even further than LLMCarbon, containing only a single parameter compared to the multi-parameter nature of polynomial approximations. Why is this single-parameter approach purported to yield higher accuracy? The authors are encouraged to offer detailed explanations or empirical validation demonstrating how and why Equation (7) leads to improved accuracy over traditional polynomial approximations.

2. **Consideration of GPU count and parallelism settings**: The paper does not discuss varying GPU counts in training configurations, appearing to assume a single-GPU setup. It also does not address different training parallelism types, such as data, tensor, pipeline, or expert parallelism, all of which may affect results depending on GPU count. Without incorporating these parallelism factors, it is unclear how OpenCarbonEval achieves greater accuracy. How does this work account for different parallelism strategies, and are there empirical results confirming its accuracy across these configurations? Additionally, Figure 4 lacks context: how many GPUs are represented, why do some GPUs exhibit smaller variance, and how many GPUs are used for training in Tables 1 and 2?

3. **Lack of model architecture information**: The study appears to consider only the number of parameters in ML models, without accounting for architecture specifics. While scaling laws suggest that architecture does not impact model accuracy, it significantly affects training throughput across various architectures (see Megatron paper: https://parsa.epfl.ch/course-info/cs723/papers/Megatron.pdf). The authors should provide empirical evidence to demonstrate that model architecture does not impact the carbon footprint of training.

4. **Dataset limitations**: The dataset used is limited and lacks comprehensive real-world data. Among the 863 entries in the provided table (https://epochai.org/data/notable-ai-models?view=table), only 176 entries include training times, 158 provide GPU counts, and only 31 report hardware utilization, leaving most entries without training times or hardware utilization data. With such limited information, how is \( f(x) \) in Equation (5) trained and validated? Furthermore, 603 of the 863 entries are classified as "likely," "speculative," or "no confidence." Does OpenCarbonEval rely on these uncertain data points for validation while claiming higher accuracy? The authors should discuss the limitations associated with the dataset quality and address the impact on the reliability of their conclusions.

---

### Official Review · Reviewer_KsLG · 2024-11-02

**Soundness:** 1
**Presentation:** 2
**Contribution:** 2
**Rating:** 3
**Confidence:** 4

**Summary:**

The paper presents OpenCarbonEval, an innovative approach for estimating the carbon footprint of training large ML models, with claims to improve prior models by incorporating hardware-specifications, embodied and operational carbon estimation, and dynamic power consumption. It introduces an α parameter to model dynamic power consumption and introduces an open-source dataset of 110 models across multiple large-scale ML tasks for validation of the proposed approach.

**Strengths:**

1.	Relevant Topic: The environmental impact of large ML models is an important concern, and OpenCarbonEval’s focus on a general framework for carbon footprint estimation. OpenCarbonEval showcases improved error rate in comparison to LLMCarbon across various large-scale ML models.

2.	Multi-Domain Scope: The method’s attempt to generalize across model types, hardware types, and tasks, potentially making it more versatile than existing carbon estimation among the estimation methods.

3.	Dataset Creation: OpenCarbonEval contributes an open resource by curating a dataset of carbon emissions containing 110 records.

**Weaknesses:**

1.	Inadequate Justification for the α Parameter: The derivation of the α parameter lacks theoretical depth, as the paper does not substantiate the choice of logarithmic modeling. Providing empirical or theoretical evidence for using f(t)=ln(1+αt) would strengthen its validity; a comparison with alternative functions could clarify this choice.
2.	Limited Model Generalization: OpenCarbonEval does not convincingly show its ability to generalize across diverse ML tasks and architectures. The adaptability of the α parameter remains unclear, particularly for models outside the initial dataset. Additional validation across a wider range of model types by extending Table 1, 2 will reinforce its versatility. Detailed results for the validation of the method is required.
3.	Lack of Explanation for Equations: The paper lacks the connection between equation (2) and equation (3), and also lacks the explanation of how the Lcomputation is used to estimate the energy consumption E. Moreover, the Clifelong needs to be elaborated in terms of how it is attained.
4.	Comparison with results for LLMCarbon: Can the authors present the analysis of same models and hardware combinations presented in Table 4 in the LLMCarbon paper?
5.	Justification or Citation for Assumption: The assumption of 1-year GPU lifespan for the embodied carbon estimation lacks justification or citation from a reliable source.
6.	Overlooked Factors in Operational Carbon Calculation: OpenCarbonEval does not account for essential factors like Power Usage Effectiveness (PUE) in data centers, leading to potential underestimations of emissions. Including PUE in calculations would create a more realistic operational carbon estimate.
7.	Simplistic Treatment of Training Dynamics: OpenCarbonEval applies Little’s Law simplistically, assuming a steady state in training dynamics, which oversimplifies the training process. More practical grounding, perhaps through empirical evidence, would enhance applicability in ML contexts. LLMCarbon addresses this by using detailed hardware efficiency and optimal parallelism settings, providing a robust framework for accurately modeling training dynamics.
8.	Embodied Carbon Calculation: OpenCarbonEval’s approach to embodied carbon appears oversimplified, lacking in-depth parameters that affect emissions, such as hardware-specific manufacturing and lifetime estimates. Moreover, the Clifelong needs to be elaborated in terms of how it is attained.

**Questions:**

Address all the questions raised in the weakness points, mentioned in the following again:

1.	Inadequate Justification for the α Parameter: The derivation of the α parameter lacks theoretical depth, as the paper does not substantiate the choice of logarithmic modeling. Providing empirical or theoretical evidence for using f(t)=ln(1+αt) would strengthen its validity; a comparison with alternative functions could clarify this choice.
2.	Limited Model Generalization: OpenCarbonEval does not convincingly show its ability to generalize across diverse ML tasks and architectures. The adaptability of the α parameter remains unclear, particularly for models outside the initial dataset. Additional validation across a wider range of model types by extending Table 1, 2 will reinforce its versatility. Detailed results for the validation of the method is required.
3.	Lack of Explanation for Equations: The paper lacks the connection between equation (2) and equation (3), and also lacks the explanation of how the Lcomputation is used to estimate the energy consumption E. Moreover, the Clifelong needs to be elaborated in terms of how it is attained.
4.	Comparison with results for LLMCarbon: Can the authors present the analysis of same models and hardware combinations presented in Table 4 in the LLMCarbon paper?
5.	Justification or Citation for Assumption: The assumption of 1-year GPU lifespan for the embodied carbon estimation lacks justification or citation from a reliable source.
6.	Overlooked Factors in Operational Carbon Calculation: OpenCarbonEval does not account for essential factors like Power Usage Effectiveness (PUE) in data centers, leading to potential underestimations of emissions. Including PUE in calculations would create a more realistic operational carbon estimate.
7.	Simplistic Treatment of Training Dynamics: OpenCarbonEval applies Little’s Law simplistically, assuming a steady state in training dynamics, which oversimplifies the training process. More practical grounding, perhaps through empirical evidence, would enhance applicability in ML contexts. LLMCarbon addresses this by using detailed hardware efficiency and optimal parallelism settings, providing a robust framework for accurately modeling training dynamics.
8.	Embodied Carbon Calculation: OpenCarbonEval’s approach to embodied carbon appears oversimplified, lacking in-depth parameters that affect emissions, such as hardware-specific manufacturing and lifetime estimates. Moreover, the Clifelong needs to be elaborated in terms of how it is attained.

OpenCarbonEval addresses a timely topic and proposes interesting methods for estimating emissions, but fundamental conceptual and methodological gaps needs to be clarified. Without rigorous validation, robust comparisons, and clearer theoretical grounding for key parameters, the method may not yet be practical for diverse ML scenarios. Addressing these weaknesses could make OpenCarbonEval a valuable contribution in the future.

---

### Official Review · Reviewer_3pyt · 2024-11-03

**Soundness:** 1
**Presentation:** 2
**Contribution:** 2
**Rating:** 3
**Confidence:** 2

**Summary:**

This paper proposes a tool named OpenCarbonEval to estimate energy consumption and carbon emissions during the training process of large ML models. The authors present a new formulation for estimating the training carbon footprint of various models and evaluate the effectiveness of their approach in comparison to related work.

**Strengths:**

- Timely problem
- Good motivation

**Weaknesses:**

- Novelty
- Soundness
- Insufficient hardware details

**Questions:**

Thank you for submitting to ICLR 2025. The research problem is both interesting and timely. I have the following questions:

- The differences between OpenCarbonEval and related works, such as CodeCarbon and Eco2AI, appear to be mainly focused on engineering efforts. For example, they could just refactor the code and adopt equation (1). Could the authors expand on their comparisons from a research perspective or discuss additional experimental results?

- The accuracy of estimating f(t) needs improvement. Could the authors clarify how they derived the function, such as comparing with other possible functions?

- Distributed deep learning systems could be complex. Models mapped onto the same class of GPUs can result in varying energy consumptions. Could the authors provide experimental results on how OpenCarbonEval accounts for differences in energy consumption when adjusting for data parallelism, tensor parallelism, pipeline parallelism, and data offloading? Additionally, how does the carbon footprint scale with the number of GPUs?

- The authors have conducted a substantial number of experiments. Are there any key takeaways from these experimental results? For instance, how does the knowledge of CO₂ emissions impact future training strategies?

---

### Note · Authors · 2024-11-26

I have read and agree with the venue's withdrawal policy on behalf of myself and my co-authors.